# Optimizing Pharmacotherapy During Implementation of Enhanced Recovery After Surgery (ERAS) in Ambulatory Urologic Oncology Surgery: Narrative Review

**DOI:** 10.3390/cancers17040614

**Published:** 2025-02-11

**Authors:** Jaret K. Shook, Thomas E. Hutson, Eric A. Singer, Saum B. Ghodoussipour

**Affiliations:** 1Heritage College of Osteopathic Medicine, Ohio University, Athens, OH 45701, USA; 2Division of Hematology and Medical Oncology, School of Medicine, Texas Tech University Health Science Center, Lubbock, TX 79430, USA; thomas.hutson@ttuhsc.edu; 3Division of Urologic Oncology, Comprehensive Cancer Center, The Ohio State University, Columbus, OH 43210, USA; eric.singer@osumc.edu; 4Section of Urologic Oncology, Rutgers Cancer Institute and Rutgers Robert Wood Johnson Medical School, New Brunswick, NJ 08901, USA; saum.ghodoussipour@rutgers.edu

**Keywords:** enhanced recovery after surgery, ERAS, ambulatory urologic surgery, surgical outcomes, urologic oncology, surgical pharmacology

## Abstract

A common goal among urologists is to perform surgery using techniques that are minimally invasive and allow for a rapid recovery. Cancer is among the many conditions that urologists seek to treat with forms of minimally invasive surgery. The surgery itself, however, is only one element of the care a patient receives during the days leading up to and following surgery. In order to improve the likelihood of patients having an uncomplicated recovery, urologists are implementing strategies called Enhanced Recovery After Surgery protocols. These protocols are implemented not only on the day of surgery, but also on the days surrounding surgery. The goal of this narrative review is to highlight the medications used to promote positive outcomes during this period. In doing so, we seek to provide a comprehensive review of the medication options applicable to the many cancer surgeries that urologists perform.

## 1. Introduction

As medical and surgical knowledge continues to advance, so too does our capability to perform procedures via less invasive approaches in ambulatory or outpatient settings. According to the Ambulatory Surgery Center Association, the most recent data show that there are 6223 Medicare-certified ambulatory surgery centers (ACSs) in the United States, performing roughly 23 million procedures a year [1]. To appreciate what it means to provide care at an ASC, it is important to highlight the requirements that these centers must meet before being certified by the Centers for Medicare and Medicaid Services. Notably, an ASC is meant to “operate exclusively for the purpose of providing surgical services to patients not requiring hospitalization and in which the expected duration of services would not exceed 24 h following an admission” [2]. Taking this into consideration, there has been an emphasis on implementing recovery protocols called Enhanced Recovery After Surgery (ERAS) pathways. ERAS is a comprehensive, multifaceted approach to the perioperative management of surgical patients that emphasizes improving recovery to promote improved surgical outcomes [3]. While the implementation of ERAS is not limited to ambulatory surgery, the protocols’ value is amplified, as the 24 h discharge time frame can be a barrier to a procedure being performed in the ambulatory setting. Focusing on the specialty of urology, advancements in techniques such as laparoscopic, robotic, and endoscopic approaches have allowed for more procedures to be performed in the ambulatory setting. As evidenced by Witherspoon and colleagues, an ambulatory urologic surgical procedure is not without risk, however, and there is room for improvements in morbidity outcomes. The authors highlighted this concept by identifying several common ambulatory procedures, including ureteric stent insertion and greenlight laser photo vaporization of the prostate, which pose a significant risk of emergency room encounters within 90 days of the procedure [4]. The acknowledgement of such risks has resulted in ERAS guidelines and developmental protocols for urologic procedures, including radical cystectomy, the transurethral resection of bladder tumors, radical prostatectomy, ureteroscopic stone treatment with stent placement, and benign prostatic hyperplasia [5,6,7,8,9].

Despite ERAS emphasizing “recovery after surgery”, ERAS does not begin at the conclusion of a procedure. Instead, ERAS is a comprehensive protocol that begins with patient optimization prior to surgery. Undoubtedly, a patient who is optimized before surgery will ultimately have a more efficient recovery and fewer complications in the perioperative period [3,5,10,11]. With this in mind, ERAS should begin with a thorough preoperative evaluation. This evaluation will help to ensure that the surgical team has a complete clinical picture and can provide all the necessary interventions to increase the chances of positive outcomes. The preoperative evaluation is not the responsibility of one provider, but instead, the entire surgical team should contribute their respective clinical expertise. In relation to ambulatory urologic oncology procedures, this may include the urologist, anesthesiologist, pharmacist, advanced practice providers, nursing staff, and others involved in the management of the patient [11]. In mentioning the members of the surgical team, it is essential to highlight the value that the pharmacist provides with regard to patient optimization. Not only is the pharmacist able to manage the patient’s chronic maintenance medications, but additionally, they serve as a resource for determining appropriate medical therapies based on the patient’s pre-existing conditions, lab parameters, the surgical procedure, the perceived risks of the operation, and postoperative complications. Thus, utilizing a pharmacist throughout the perioperative period will contribute to improved patient outcomes [12]. In this narrative review, we summarize the implementation of pharmacotherapy in the setting of ERAS for ambulatory urologic oncology surgery.

## 2. Methods

Using the databases Academic Search Complete, Access Medicine, ClinicalKey, and PubMed, we performed a non-systematic review of articles from January 2005 to December 2024. The terms used to generate the search included the following: “ERAS”, “enhanced recovery after surgery”, “urologic surgery”, “urology”, “ambulatory urologic surgery”, “urologic oncology”, “urologic oncology surgery”, “optimization”, “pharmacotherapy”, “perioperative management”, “analgesia”, “antiemetics”, “bladder spasms”, “urinary retention”, “VTE prophylaxis”, “surgical outcomes”, and “surgical pharmacology”. Articles were then reviewed to identify primary and secondary sources of research that evaluated and/or reviewed one or more of the following themes: pharmacotherapy implementation in ERAS protocols, ERAS protocols for urologic disease states, and individual components of perioperative care essential to urologic surgery. These individual components included patient education, the optimization of chronic medical conditions, perioperative nutrition, frailty mitigation, antimicrobial prophylaxis, pain management, nausea and emesis, bladder spasms, stent discomfort, urinary retention, and venous thromboembolism (VTE) prophylaxis. The inclusion criteria for primary and secondary research included being an original, full-text article from a peer-reviewed journal that was published in January 2005 or onward with a focus on perioperative care. The exclusion criteria included articles not available online or published in a language other than English. Clinicaltrials.gov was used to collect information on the ongoing clinical trials cited in this review. Tertiary sources further supplemented the development of this narrative review. Specifically, pharmacotherapy text sources were used to develop the pharmacotherapy tables included in the review. Additionally, the statistics and definitions crucial to understanding the practice of ambulatory surgery were gathered from the referenced web resources.

## 3. Artificial Intelligence and ERAS

Given the ever-intertwining relationship between medicine and technology, it would be an injustice to discuss patient optimization and enhanced surgical recovery without discussing artificial intelligence (AI). To date, one of the most promising aspects of AI in surgery is its capacity to synthesize and analyze patient data in an effort to aid in clinical decision making. In one such example, Henderson and colleagues assessed the capability of machine learning to predict the suitability of patients for surgery at an ASC [13]. In their study, the authors provided the model with patient data including sex, age, body mass index, blood pressure, hemoglobin A1C, serum creatinine, the diagnosis codes on the patient’s problem list, the procedure being performed, if the surgeon had requested an ASC, if the patient had undergone recent surgery at an ASC, and if the patient had any previous complications from anesthesia. The authors found that “on held-out test data, the review and location models had areas under the receiver operator curves of 0.846 (95% CI, 0.837–0.855) and 0.965 (95% CI, 0.961–0.968), respectively”. Thus, the study demonstrated that machine learning has a growing capacity to stratify patients based on perceived risk and reduce the workload associated with manual reviews [13]. With such promise, however, must come the acknowledgement that AI programs will not be universally adopted due to a variety of logistical hurdles. Notably, healthcare systems will need to address the financial cost of implementing AI programs in a systematic way to ensure consistency and continuity across systems. Furthermore, this financial expenditure is likely to disproportionally impact smaller health systems, thereby increasing the disparity in services that particular systems are able to provide. Aside from the financial considerations, healthcare systems will also have to determine how to effectively train healthcare staff who are already time-constrained. Concern can also be drawn to the reality that there are multiple AI software programs available. If there is not a systemwide requirement for a given AI program, clinicians may receive different evaluations of the same question. Such logistical considerations will likely necessitate feasibility and cost analysis studies moving forward. Finally, while the aforementioned study serves as a blueprint for the effective implementation of AI in presurgical evaluations, further research will help to determine the capacity that AI has to differentiate clinically significant and insignificant patient data. As these concerns are addressed in time, the pairing of ERAS protocols with AI-driven risk stratification will aid urologists in ensuring patient optimization throughout the perioperative period.

## 4. Components of ERAS

### 4.1. Optimization

For appropriate patient optimization, urologic perioperative care should emphasize patient education, risk stratification, the stabilization of chronic medical conditions, perioperative nutrition, frailty mitigation, antimicrobial prophylaxis, pain management, nausea and emesis control, bladder spasm control, reducing stent discomfort, urinary retention prevention, and VTE prophylaxis. Optimization cannot be comprehensive until each of these components is addressed. This also implies that each component can also serve as a potential barrier. The acknowledgment of these components lays the foundation for the discussion of how each component can be addressed in the preoperative, intraoperative, and postoperative phases of patient care.

### 4.2. Patient and Caregiver Education

In beginning the discussion of the individual components of ERAS, none should take greater precedence than patient and caregiver education. It is well-accepted that a patient who is well-informed regarding their care contributes positively to recovery. Similar to preoperative assessment, patient and caregiver education should take place in a team-based fashion. It is vital that education does not occur as one isolated event, but instead is reiterated both pre- and postoperatively. Preoperatively, a comprehensive discussion should begin at the patient’s preoperative office visit. This allows time to be thorough while also giving the patient a chance to consider any further questions that may arise after the visit. During this initial discussion, it is appropriate to review the preoperative diagnosis, surgical approach, application of anesthesia, incision site, potential need for catheters, drains, or stomas, risk of complications, expectant management of pain, nausea, bladder spasms, stent discomfort, and urinary retention, VTE prophylaxis, and a recovery timeline [11]. Furthermore, in an effort to ensure the accuracy of patient history, a thorough review of the patient’s past medical and surgical history, as well as a medication reconciliation, is necessary. These individualized histories are paramount to the perioperative management of a surgical patient. Postoperatively, another review of management considerations should be provided to caregivers, while also scheduling a close follow-up office visit with the patient. At each of these encounters, it is appropriate to provide a comprehension check to ensure that the patient and caregiver have a thorough understanding. Commonly, a “teach back” method is used for this assessment [11].

### 4.3. Assessment of Chronic Medical Conditions

As eluded to above, an assessment of patients’ chronic medical conditions should take high priority in the preoperative evaluation. With this assessment, the urologist is able to assess a patient’s suitability for surgery and gather insights into any potential risk factors and complications that may arise [11]. Conditions of particularly high importance to evaluate include anemia, malnutrition, pulmonary conditions including obstructive sleep apnea, asthma, and COPD, cardiac conditions including coronary artery disease, hypertension, and atrial fibrillation, and chronic kidney disease and diabetes [5,11,12]. At this juncture in preoperative care, the pharmacist can assist the urologist in optimizing any of the conditions listed above with appropriate pharmacotherapy [12]. In assessing anemia, evaluating a complete blood count is helpful in determining whether supplementation with iron, folate, vitamin B12, or erythropoietin may be necessary. Keys to the optimization of common chronic respiratory conditions include smoking cessation for two to four weeks preoperatively, in addition to adherence to maintenance inhalers and the use of CPAP for patients with obstructive sleep apnea [11]. The preoperative management of cardiac conditions may include a cardiac work-up involving stress testing, electrocardiograms, and echocardiograms, if appropriate, as well as adherence to prescribed pharmacotherapy for blood pressure and arrythmia control. For patients on long-term anticoagulation, a discussion regarding the duration to hold anticoagulants should occur in conjunction with the patient’s cardiologist [14]. The preoperative management of chronic kidney disease may vary depending on the disease progression. While early-stage disease may necessitate dose adjustments to the patient’s medication regimen, late-stage disease may involve timing dialysis around the planned procedure or the avoidance of particular pharmacotherapy altogether. Finally, diabetes management cannot go understated in the perioperative window. Given that elevated blood glucose can give rise to multiple complications, including impaired wound healing and increased infection risk, glucose control and monitoring are essential to improve surgical outcomes [11]. It should be noted, however, that the increasingly popular GLP-1 receptor agonists do carry an increased risk for the aspiration of gastric contents under anesthesia through the mechanism of delayed gastric emptying, and should be addressed accordingly with the anesthesiologist [15].

### 4.4. Perioperative Nutrition Management

Historically, recommendations regarding perioperative nutrition have primarily focused on carbohydrate loading and preoperative fasting. Given that surgery is known to increase a patient’s metabolic requirements, addressing malnutrition is important in improving surgical outcomes and promoting a faster recovery. A key strategy that has been adopted in surgery involves carbohydrate loading in the days leading up to surgery prior to presurgical fasting. This strategy helps to reduce insulin resistance and maintain lean muscle mass and strength in the perioperative period [5,11].

At present, two phase III trials are currently being conducted to further quantify the effects of immunonutrition on the outcomes of surgical patients. Conceptually, immunonutrition is focused on modulating the activity of the immune system by providing specific nutrients. The INCyst trial, conducted by Ilaria Lucca and colleagues, aims to determine “the impact that immunonutrition has on post-operative complications in patients undergoing a cystectomy”. The researchers intend to preoperatively provide an oral solution containing arginine, ribonucleic acid, and omega-3 polyunsaturated fatty acids and determine its impacts on “nutritional status, immunological function and clinical outcomes of surgical patients” using infectious complications, mortality rates, and complication-free survival rates as quantifiable outcomes [16]. Similarly, Hamilton-Reeves and colleagues are conducting research on “how well nutrition therapy works in improving immune system in patients with bladder cancer that can be removed by surgery.” In the study, patients will preoperatively receive immune-modulating oral solutions and will be monitored postoperatively for several outcomes, including postoperative complications, postoperative infections, and disease-free survival, among others [17]. Moving forward, further evidence to support immunonutrition’s impact on surgical outcomes will need to be presented before widespread adoption is seen. This may include addressing patient adherence to the regimen, cost–benefit analysis, and postoperative outcomes.

In addition to these new developments, fasting recommendations are beginning to shift with the acknowledgment that prolonged fasting decreases glycogen stores, while increasing inflammation and contributing to losses in lean mass [11]. In a separate study from the one described above, Hamilton-Reeves and colleagues are conducting a study that aims to investigate this concept further. The “Carbohydrate Ingestion Prior to Surgery (CIPS)” study has been designed to “determine the impact of taking a specialized form of carbohydrate in the immediate preoperative period on metabolic markers, surgical outcomes and patient health” [18]. With this evolving research, the urologist and anesthesiologist must strike a balance between minimizing aspiration risk while acknowledging the impact that prolonged fasts have on patient physiology.

### 4.5. Frailty Mitigation

In line with the aforementioned topics of chronic medical conditions and perioperative nutrition, addressing frailty in the operative patient is necessary to promote improved surgical outcomes. Despite providers being able to picture a “frail” patient in their mind, there tends to be a lack of consensus on the clinical definition of frailty. In an effort to mitigate variation, the Geriatric Advisory Panel of the International Academy of Nutrition and Aging set out to define frailty. The group suggested that frailty should be considered as a “predisability state”. That is to say that “frail” patients are “in a state of increased vulnerability to stressors” [19]. Building upon this definition, various criteria, scores, indexes, and questionnaires have been introduced to categorize and quantify frailty [20,21,22]. One such index was created by Rockwood and colleagues. The authors developed a seven-point Clinical Frailty Scale which categorized 2305 patients aged 65 and older participating in stage 3 of the Canadian Study of Health and Aging into “very fit”, “well”, “well, with treated comorbid disease”, “apparently vulnerable”, “mildly frail”, “moderately frail”, and “severely frail”. This categorization occurred on the basis of a clinical evaluation of the patients’ activity and independence level. After categorization, these patients were subsequently followed in a 5-year prospective cohort study. Data analysis of the categorized patients highlighted that, with each one-category worsening of frailty status, there was a statistically significant increase in death at 70 months (21.2%, 95% CI 12.5–30.6%), as well as an increased rate of entry into institutional care at 70 months (23.9%, 95% CI 8.8–41.2%) [20].

In urologic oncology surgery patients, similar trends correlating frailty status and surgical outcomes have been documented. In one such case, Savin and colleagues developed an 11-item modified frailty index (mFI) and performed a retrospective analysis correlating the modified frailty scores of 292 patients with their outcomes following radical cystectomy [21]. The presence of 11 conditions in a patient’s medical history contributed to the index. These included diabetes mellitus, non-independent functional status, pulmonary disease, heart failure, hypertension, peripheral vascular disease, impaired sensorium, cerebrovascular accident, and neurologic deficit. Each condition contributed 1 point, and the categorization followed as such, “robust” if the mFI was 0, “pre-frail” if the mFI was 1–2, and “frail” if the mFI was equal to or greater than 3. Furthermore, patients with an mFI of 0–1 were considered to be “low risk” and patients with an mFI of 2 or greater were “high risk”. In the authors’ analysis, it was shown that overall survival and cancer-specific survival were “significantly lower for patients with an mFI score of equal or greater than 2 (*p* = 0.007 and *p* = 0.03, respectively)” [21]. Considering the two previous studies, it becomes evident that frailty should be a priority in a urologist’s assessment of a patient’s surgical risk.

Given that frailty itself often presents as a byproduct of multiple comorbidities, a urologist seeking to address frailty in a perioperative patient should take a comprehensive approach. Such an approach, however, is likely to necessitate the need for comanagement with other providers. Here, system-based specialists, geriatricians, and pharmacists can all provide value. In an effort to determine the effect of geriatric comanagement on the postoperative mortality of geriatric patients with cancer, Shahrokni and colleagues conducted a retrospective cohort study of 1892 patients, all of whom were 75 years of age or older [23]. The patients were separated into two groups, those who received geriatric comanagement perioperatively or those who were managed by the surgical service alone. Patients receiving perioperative care through the geriatrics service received additional interventions both pre- and postoperatively. Additional preoperative interventions included further consultations as deemed necessary by the geriatrics team and geriatric-focused patient education regarding surgical preparation and recovery. Postoperative interventions for the geriatric comanaged group included postoperative follow-up by the geriatrics service on postoperative days one through three, assistance with the reintroduction of medications, delirium mitigation strategies, encouraging early mobility and activity, education on incentive spirometry, deep vein thrombosis prophylaxis, pain management, bowel regimen management, and hospital disposition planning. Multivariate logistic regression analysis showed that “patients in the geriatric comanagement group were less likely to die within 90 days after surgical treatment (odds ratio [OR], 0.43 [95% CI, 0.28–0.67]; *p* < 0.001)” [23]. Taking these data into consideration in conjunction with the pathophysiology associated with frailty, it becomes apparent that a urologist would be wise to address frailty perioperatively with the help of an intradisciplinary team.

### 4.6. Antimicrobial Prophylaxis

Although the time course of surgical site infections typically extends beyond the 24 h window associated with ambulatory surgical procedures, antimicrobial prophylaxis is an important consideration in ERAS. The consensus among both the CDC and AUA is that a single dose of a parenteral antimicrobial needs to be infused prior to skin incision, with administration typically occurring within 1 h preceding the procedure [24,25,26]. Often, a first-generation cephalosporin provides sufficient prophylactic coverage, but the urologist and pharmacist should work together to determine if other antimicrobials are necessary based on procedure risk, local antibiograms, and patient factors such as renal function [12,24]. It is important to note that the CDC and AUA recommend against the postoperative continuation of prophylactic antimicrobials, given the risk of antimicrobial resistance and the chance for opportunistic infections with prolonged antibiotic use [25,26].

### 4.7. Pain Control

Despite opioid analgesics being the historical standard for pain control in surgery, the ongoing opioid epidemic has heightened the need to promote multimodal opioid-sparing pain regimens. With this in mind, there is value in research surrounding ERAS protocols and their effect on perioperative opioid usage. In a study conducted by Ghodoussipour and colleagues, the authors sought to evaluate how ERAS protocols adopted for radical cystectomy effected postoperative outcomes in patients with preoperative narcotic use when compared to opioid-naïve patients [27]. In a retrospective analysis, 34 patients with preoperative opioid use were matched to 68 opioid-naïve patients from a prospectively maintained bladder cancer database. Preoperative opioid use was defined as having an active prescription for narcotic pain medication for at least 30 days prior to surgery. Each of these patients had undergone an open radial cystectomy with pelvic lymph node dissection and urinary diversion for urothelial carcinoma with intent to cure and were enrolled in an institutional ERAS protocol. The pain management outlined in this protocol consisted of the “intraoperative use of intravenous acetaminophen and ketorolac, postoperative around-the-clock oral acetaminophen, IV ketorolac, and infusion of ropivacaine via para-incisional subfascial catheters”, with narcotic use being reserved for breakthrough pain. The results of the data analysis showed that preoperative narcotic users had higher postoperative pain scores per postoperative day and received more opioids per postoperative day when compared to opioid-naïve patients. However, it was concluded that there was no significant difference in the median length of stay (4 days vs. 4 days, *p* = 0.6) or rate of complications at 30 days (64.7% vs. 60.3%, *p* = 0.8) and 90 days (82% vs. 75%, *p* = 0.4). Notably, this included no difference in GI complications (29.4% vs. 26.5%, *p* = 0.8) or postoperative ileus (11.8% vs. 20.6%, *p* = 0.2). Both groups also had similar re-admission rates at 30 days (20.6% vs. 13.2%, *p* = 0.3), while previous opioid users had a higher re-admission rate at 90 days (41.2% vs. 20.6%, *p* = 0.03). Among the authors’ conclusions, the lack of significant differences in GI complications and postoperative ileus had a high clinical significance. This was due to the fact that an increased length of stay and postoperative ileus are associated with higher levels of postoperative opioid use [27]. In applying these findings to ERAS in ambulatory urologic oncology surgery, it should be taken into account that this opioid-sparing regimen did not result in a significant difference in length of stay or complication rate, regardless of the status of the patients’ preoperative narcotic use.

Invariably, opioids will continue to serve a purpose in the context of perioperative pain control. Evidence, such as that described above, however, highlights the ability to achieve equivocal outcomes while minimizing their use. An additional study conducted by Xu and colleagues provided reaffirming results surrounding the efficacy of opioid-sparing regimens as part of an ERAS protocol following urologic procedures [28]. In the study, the authors retrospectively compared the use of opioids in traditional protocols compared to ERAS protocols. Their findings described that, while patients in the enhanced protocol reported more pain on a visual analog scale (3.1 vs. 1.14, *p* < 0.001), there was “significantly less opioids per day (4.9 mg vs. 20.67 mg morphine equivalents, *p* < 0.001).” Furthermore, there was a decreased incidence of postoperative ileus (7.3% vs. 22.2%, *p* = 0.003) and a shorter median length of hospital stay (4 vs. 8 days, *p* < 0.001) [28].

Notably, the opioid-sparing pain regimens implemented across varying ERAS protocols can differ from one to another. Most often, these protocols involve a combination of pre-emptive analgesia, nerve blocks, opioid-sparing intraoperative and postoperative pain regimens, and non-pharmacological pain treatments. Non-opioid options commonly seen for pre-emptive use include acetaminophen, NSAIDs, gabapentin, pregabalin, ketamine, and local anesthetics [24,29,30]. In the setting of urologic surgery, the use of epidural analgesia, intrathecal morphine (ITM), and transverse abdominal plane (TAP) blocks is also implemented preoperatively [24]. Intraoperatively, anesthesiologists have been able to implement opiate-sparing or opiate-free anesthesia techniques to minimize overall exposure. Smith and colleagues demonstrated an opiate-free technique in urologic procedures that also reduced the postoperative dosing of opiates, with statistically significant results. The regimen included acetaminophen, ibuprofen, midazolam, dexmedetomidine, lidocaine, bupivacaine, and volatile anesthetics [31]. Postoperative options include the judicious use of opioids, as well as acetaminophen, NSAIDs, gabapentin, pregabalin, and ketamine [29,30,32,33]. Each of the previously mentioned analgesic agents and their necessary considerations are further highlighted in Table 1 below. With such variations in the pain management component of ERAS protocols, there is an obvious need for continued research. This may include head-to-head evaluations of individual ERAS protocols regarding their effectiveness or seek to address variations between protocols in dosage, timing, or route of administration. Of note, the non-pharmacologic pain treatments recommended in the 2016 Pain Guidelines include transcutaneous electrical nerve stimulation, acupuncture, and cognitive behavioral therapies, but these treatment modalities play a more significant role in long-term pain control [32,33].

### 4.8. Antiemetics

While pain is an ever-persistent element of patients’ postoperative presentations, postoperative nausea and vomiting (PONV) can vary in duration, onset, and severity. These variations can be attributed to patient risk factors such as prior history of PONV, being a nonsmoker, being female, having a young age, the use of inhalation anesthetics, and the use of perioperative opioids [5,24,34,35]. Nonetheless, the patients whom PONV impacts endure substantial discomfort, as well an obstacle that may preclude discharge. In order to combat this common postoperative complication, the urologist and surgical team must deploy a multimodal strategy similar to that seen in pain control. When necessary, this strategy should be centered around antiemetic prophylaxis as opposed to reactionary treatment for the nausea and vomiting. Whether or not to use prophylactic treatment should be based on an assessment of the number of risk factors that a patient demonstrates [35]. Due to the fact that PONV is mechanistically complex, when prophylaxis is appropriate, pharmacologic agents with differing mechanisms of action, described in Table 2, should be prescribed in conjunction with altered aspects of perioperative management [34,35]. To date, 5-HT3 receptor antagonists (ondansetron, granisetron, and palonosetron) have shown significant efficacy for PONV prophylaxis. These agents are provided at the end of surgery, but prior to symptom onset. Dexamethasone has also been shown to be effective, especially when used in conjunction with 5-HT3 receptor antagonists. Neurokinin 1 receptor antagonists are another option for PONV and provide the benefit of a prolonged duration of action. While scopolamine and metoclopramide can be used for PONV, their use is usually reserved due to their anticholinergic and extrapyramidal side effect profiles, respectively [24,29,30,34,35]. In addition to pharmacologic therapies, the adjunct antiemetic therapies commonly used include minimizing opioid exposure, nitrous-oxide-sparing anesthesia, acupressure of the pericardium 6 acupoint, and ensuring adequate intraoperative hydration [24,34,35].

### 4.9. Bladder Spasms and Stent Discomfort

Throughout urologic procedures, catheters, drains, and stents are commonly introduced to aid in natural urine flow, monitor urinary output, resolve potential areas of obstruction, or promote postoperative drainage [32]. These catheters are often crucial in ensuring that acute urinary retention does not develop postoperatively. With the introduction of such devices, however, comes the potential for discomfort due to reflexive spasms in the ureters, bladder, or urethra. These spasms can present a challenge to the urologist, because a balance must be struck between promoting appropriate flow and managing the associated pain. Unfortunately, either of these results can preclude timely discharge or impact complication rates in the perioperative period. In order to overcome this challenge, the urologist should consider the implementation of pharmacologic therapy to reduce spasms while maintaining the catheters, drains, and stents as necessary. To address these spasms, current pharmacologic therapy options, highlighted in Table 3, include antimuscarinic receptor antagonists, intravesical bupivacaine injections, and alpha-1 adrenergic receptor antagonists, as well as multimodal pain control [29,30,36]. Of the agents listed above, oxybutynin and other anticholinergics in the class must be used cautiously due to their associated anticholinergic risks. One of the risks of greatest concern is cognitive slowing in the geriatric patient population, leading to an increased risk for falls and delirium. In an effort to provide further pharmacologic options, Tae and colleagues attempted to demonstrate that mirabegron, a beta-3 receptor agonist, could provide relief for bladder spasms. The authors hypothesized that mirabegron may be efficacious based on the physiologic similarities of bladder spasms to overactive bladder [36]. In the study, 100 patients undergoing ureteric stent placement were randomized to receive no treatment or mirabegron during the stenting period. Their findings showed that Ureteral Stent Symptom Questionnaire (USSQ) body pain scores (21.96 vs. 13.96, *p* = 0.007), overall pain scores (5.58 vs. 2.83, *p* = 0.002), and the duration of analgesic use in days (5.5 vs. 2.6, *p* = 0.006) were lower among the mirabegron treatment group. There was found to be no significant difference in USSQ urinary symptom scores (32.58 vs. 27.92, *p* = 0.582), however. While providing promising results, the authors concluded that more extensive confirmatory studies are needed before the widespread implementation of perioperative mirabegron should be pursued [36].

### 4.10. Urinary Retention

Urinary retention can also be a cause for concern following urologic procedures. Traditionally, in order to combat postoperative urinary retention and monitor urinary output and hematuria, catheters are placed via a transurethral approach and then withdrawn in the days following surgery. Suprapubic catheters can also be placed, but are a more permanent dwelling catheter. While catheters do provide such benefits and are essential in postoperative care, the risk of catheter-associated urinary tract infections (CAUTI) is a driving reason to remove catheters in a timely manner [32]. Furthermore, catheter placement in the operating room following TURBT has been shown to be associated with an increased incidence of gross hematuria, resulting in unplanned hospital returns [37]. In an effort to reduce the duration of postoperative catheter use and facilitate early catheter removal without increasing the risk of urinary retention, the urologist should consider the use of pharmacologic therapy. In particular, alpha-1 receptor antagonists and the muscarinic agonist bethanechol, outlined in Table 4, have shown pharmacologic effects in reducing the incidence of urinary retention [29,30]. One example demonstrating tamsulosin’s efficacy can be seen in the study by Jeong and colleagues. The researchers were able to demonstrate that patients treated with tamsulosin in the perioperative period (from 1 day prior to surgery to 14 days following surgery) following robot-assisted laparoscopic radical prostatectomy had statistically significant lower rates of acute urinary retention when compared to a placebo group (7.3% vs. 17.4%, *p* = 0.018) [38]. Of note, 5 alpha-reductase inhibitors have shown the ability to relieve urinary retention through a reduction in prostatic mass, however, meaningful effects have not been achieved in the context of ambulatory surgery, as therapy deprives the prostate of dihydrotestosterone (DHT) over a period of months [29,30].

### 4.11. VTE Prophylaxis

It is well-demonstrated that surgery is a risk factor for VTE. Furthermore, if there is a superimposed malignancy, there is an additive effect to the risk for VTE. Accounting for the range of invasiveness of urologic procedures performed, a risk assessment is necessary to determine the extent of the pharmacologic or mechanical VTE prophylaxis required. While it is essential to prevent thromboembolism, it is equally important to reduce the risk of bleeding associated with a given procedure. Calculating a Caprini score is a validated tool for determining the risk of a VTE event occurring [39]. Once a risk assessment is complete, if a patient is deemed low-risk for VTE, mechanical prophylaxis is preferred over pharmacologic. For patients in the moderate- or high-risk categories, pharmacologic prophylaxis is recommended if there is not high bleeding risk [32,39]. In the perioperative setting, the pharmacologic therapies of choice include heparin and low-molecular-weight heparin (LMWH), as described in Table 5. While heparin can form greater quantities of the heparin–antithrombin quaternary complex due to its size and inactivate more thrombin, LMWH is generally preferred, owing to its more consistent bioavailability and longer half-life. The recommended dosing of heparin for patients requiring pharmacologic VTE prophylaxis is 5000 units subcutaneously every 8 h. For LMWH, the recommended dosing for VTE prophylaxis is 40 mg subcutaneously daily. It is worth noting, however, that renal dose adjustments must be taken into account for LMWH when CrCl falls below 30 mg/dL [29,30]. Regarding the duration of postoperative therapy, in patients at a moderate to high risk of VTE without a high bleeding risk, 1 week of prophylaxis is recommended. This recommendation changes when these individuals are undergoing surgery for cancer, at which time, the recommended duration of therapy becomes 4 weeks [39].

## 5. Conclusions

Within urologic oncology surgery, continuing to develop and validate ERAS protocols will provide added benefits to the outcomes of surgical patients. This will take the form of an increasing capacity to perform advanced urologic oncology procedures in the ambulatory surgery setting, due in large part to patient optimization in the perioperative period and minimally invasive techniques. Protocols such as the one developed by Rezaee and colleagues for the transurethral resection of bladder tumors serve as a great representation. With the development of their protocol, the authors hope to implement ERAS into an extremely common ambulatory procedure and provide significant improvements in recovery quality and pain control [6]. In addition to the evidence supporting the positive clinical outcomes from ERAS protocols, there has been a concurrent effort to validate the financial benefits of ERAS. This effort is largely driven by the growing financial burden associated with cancer treatment. In fact, current estimates suggest that the cost of cancer care in the United States will reach USD 246 billion by 2030 [40]. While surgical interventions only contribute to a percentage of overall costs, a surgical procedure is not an insignificant portion of the total cost of care. As a result of this medical spending, patients receiving cancer care experience debt at higher rates than the average population, are more likely to file bankruptcy, and are more likely to experience foreclosures, collections, or repossessions. Termed “financial toxicity”, urologists and surgical teams must be cognizant of the financial consequences that result from particular treatment strategies, including surgical interventions [41]. Nabhani and colleagues highlighted that the benefits of ERAS extend beyond clinical significance and into these financial considerations for patients undergoing radical cystectomy. In analyzing 201 patients (99 under standard protocol and 102 under ERAS protocol), it was found that the average 30-day costs for radical cystectomy were USD 4488 less per procedure when ERAS was implemented (USD 31,139 vs. USD 26,650, *p* < 0.0001) [42]. Further evidence to support these positive financial impacts was highlighted in a systematic review by Brooks and colleagues, which analyzed the quality of life and cost-effectiveness of ERAS for patient’s undergoing urologic oncology surgery. In the study, the authors concluded that the decreased costs associated with ERAS were associated with a decreased length of hospital stay [43].

It is necessary to keep in mind that, in order to produce these clinical, financial, and quality of life benefits, each component of patient optimization must be addressed. These components previously discussed include patient education, chronic medical conditions, perioperative nutrition, frailty mitigation, antimicrobial prophylaxis, pain control, antiemetics, bladder spasm control and minimizing stent discomfort, urinary retention prevention, and VTE prophylaxis. Table 6, below, provides a summation of these perioperative pharmacotherapy considerations in the setting of commonly encountered chronic medical conditions. In addition to these intraoperative and postoperative components, a thorough preoperative assessment is imperative to improving surgical outcomes. This assessment should be comprehensive and include an interdisciplinary healthcare team, including the urologist, anesthesiologist, pharmacist, advanced practice providers, nursing staff, and others. In light of the continuous advancements in machine learning and artificial intelligence, the potential benefits that these systems provide cannot be ignored. Not only does AI have the potential to aid in the preoperative assessment itself, but in doing so, it will potentially increase work efficiency and productivity for the team. As advancements in urologic surgical techniques continue to allow for more surgical cases in the ambulatory setting, implementing ERAS protocols will become ever more imperative. Continued research on the aforementioned components of ERAS will ultimately create cumulative protocols capable of effectively addressing each perioperative patient. It is through such protocols that recovery will continue to advance in lockstep with the procedures performed, and the ultimate goal of continuously improving patient outcomes will occur.

## Figures and Tables

**Table 1 cancers-17-00614-t001:** Pharmacotherapy options for analgesia in ERAS [29,30].

Drug	Class	Mechanism of Action	Aspects of Therapy Impacting ERAS
Fentanyl Hydrocodone Hydromorphone Morphine Oxycodone	Opioid	Strong mu-opiate receptor agonist	- Effective for moderate to severe pain - Risk of respiratory depression - Side effects include nausea, vomiting, constipation, cognitive slowing, and drowsiness - Inhibition of the urinary voiding reflex - Increase in external urethral sphincter tone
Tramadol	Opioid	Weak mu-opiate receptor agonist Inhibition of reuptake of norepinephrine and serotonin	- Effective for mild to moderate pain - Side effects include nausea, vomiting, cognitive slowing, and drowsiness - Less severe development of constipation - Can exacerbate seizure risk - Risk of serotonin syndrome
Celecoxib Ibuprofen Meloxicam Naproxen	Non-steroidal anti-inflammatory drug	Competitively inhibits both COX-1 and COX-2, thus inhibiting prostaglandin production Celecoxib and Meloxicam provide COX-2 selective inhibition	- Most effective for low to moderate pain of inflammatory origin - Side effects include dyspepsia, abdominal pain, nausea, ulcerations, and diarrhea - Salt and water retention - Acute kidney injury risk - Inhibition of platelet function - Important to hold preoperatively - Not removed by hemodialysis due to extensive protein binding
Acetaminophen	Nonopioid analgesic	Inhibits COX-1 and COX-2 at an allosteric site separate from NSAIDs, thus providing a smaller magnitude of inhibitory activity, but lacks anti-inflammatory action	- Effective for low to moderate pain - Alternative for patients in which NSAIDs are not appropriate - Well-tolerated with minimal side effects when dosed appropriately - Hepatic toxicity is the greatest concern - Caution with toxicity, since often found in combination medications such as with opioids
Gabapentin Pregabalin	GABA modulators	Modulates neuronal calcium currents through binding to the alpha-2-delta subunit of calcium channels	- Most effective for neuropathic pain - Side effects include dizziness, somnolence, and peripheral edema - Renal function must be evaluated
Bupivacaine Lidocaine	Local anesthetic	Diminishes nerve membrane permeability to sodium to block nerve conduction	- When injected, provides anesthesia to subcutaneous structures - Epinephrine is commonly added to extend the duration of action by decreasing the rate of absorption - When injected in a localized area, the risk of systemic toxicity is minimized - Systemic risk includes CNS stimulation, altered cardiac electric conduction
Bupivacaine Lidocaine Mepivacaine Ropivacaine	Neuraxial anesthetic	Diminishes nerve membrane permeability to sodium to block nerve conduction	- Neuraxial anesthesia allows for a greater area of anesthesia without drastic increases in dose that would cause concern for system toxicity - Also provides skeletal muscle relaxation around the site of injection and anesthesia of multiple nerves in the associated nerve plexus - Goal is to inject the anesthetic solution as close to the nerve bundle as possible without puncture to ensure no nerve damage occurs
Ketamine	NMDA receptor antagonist	NMDA receptor antagonism which blocks glutamate transmission	- Can be used as an anesthetizing agent, but also provides analgesic effect and anterograde amnesia - Useful for patients at risk of hypotension - Shown to reduce the development of tolerance to opioids - Used as part of a multimodal regimen to reduce opioid dosing

**Table 2 cancers-17-00614-t002:** Pharmacotherapy options for antiemesis in ERAS [29,30].

Drug	Class	Mechanism of Action	Aspects of Therapy Impacting ERAS
Dolasetron Granisetron Ondansetron Palonosetron	5-HT3 receptor antagonist	Antagonism of the 5-HT3 receptor in the chemoreceptor trigger zone, as well as vagal afferents and the area postrema Palonosetron has a greater receptor affinity and a longer half-life than first-generation agents	- Most effective for chemotherapy-induced and postoperative nausea and vomiting - Largely ineffective for delayed and anticipatory nausea and vomiting - Side effects include dizziness, constipation, fatigue, and headache - Risk of QT prolongation in first-generation agents
Aprepitant Fosaprepitant rolapitant	Neurokinin-1 receptor antagonists	Antagonism of the NK1 receptor (receptor of substance P)	- Provide the greatest efficacy for delayed nausea and vomiting associated with chemotherapy - Side effects include fatigue, constipation, and hiccups - Risk of neutropenia throughout the class of agents
Dexamethasone	Corticosteroid	Binds to corticosteroid receptor proteins and regulates expression of steroid-responsive genes	- Most effective in chemotherapy-induced nausea and vomiting - Short-term use limits the potential for the side effect profile seen with long-term corticosteroid therapies such as hyperglycemia, myopathy, immunosuppression, osteoporosis, and behavioral changes
Scopolamine	Anticholinergic	Muscarinic receptor antagonist	- Most common dosage form is transdermal patch - Most effective in motion sickness, however, some activity in postoperative nausea and vomiting - Side effects present due to anticholinergic effect including dry mouth, visual disturbances, drowsiness, and cognitive delay
Metoclopramide	Dopamine antagonist	Antagonism of the D2 receptor Antagonism of central and vagal 5-HT3 receptor Agonism of 5-HT4 receptor	- Effective in promoting GI motility and thus reducing nausea and vomiting - Can be used for chemotherapy-induced nausea and vomiting - Duration of therapy should be limited to 12 weeks or less due to the risk of extrapyramidal effects

**Table 3 cancers-17-00614-t003:** Pharmacotherapy options for bladder spasms in ERAS [29,30].

Drug	Class	Mechanism of Action	Aspects of Therapy Impacting ERAS
Darifenacin Oxybutynin Solifenacin Tolterodine Trospium	Muscarinic receptor antagonists	Antagonism of the muscarinic receptors in the bladder to lower intravesicular pressure and reduce the frequency of contractions	- Side effects include the antimuscarinic profile such as xerostomia, blurred vision, constipation, drowsiness, and dizziness - Oxybutynin is associated with the highest rate of anticholinergic effects - Solifenacin and Darifenacin show more selectivity for the M3 receptor, which limits side effect profile
Intravesicalar bupivacaine	Local anesthetic	Diminishes nerve membrane permeability to sodium to block nerve conduction, thus inhibiting nociceptors in the bladder submucosa and muscular layer	- Chemical structure promotes prolonged duration of action - Pain on administration - Minimizes systemic toxicity risk given the nature of localized anesthetics - The risk of cardiotoxicity is higher with bupivacaine as opposed to other local anesthetics like lidocaine
Mirabegron	Beta 3 receptor agonist	Agonism of the B3 receptor which is expressed in the detrusor muscle of the bladder	- Results in detrusor muscle relaxation and increased bladder capacity - Side effects include hypertension, risk of urinary tract infections, and headache

**Table 4 cancers-17-00614-t004:** Pharmacotherapy options for urinary retention in ERAS [29,30].

Drug	Class	Mechanism of Action	Aspects of Therapy Impacting ERAS
Alfuzosin Doxazosin Prazosin Silodosin Tamsulosin Terazosin	Alpha-1 adrenergic receptor antagonist	Selective antagonism of the Alpha-1 adrenergic receptor	- Alpha-1 adrenergic receptors are abundant in the prostate, prostatic capsule, prostatic urethra, and bladder neck - “First dose effect”: postural hypotension and syncope after initial dose
Bethanechol	Muscarinic receptor agonist	Agonist at the muscarinic receptors of the genitourinary tract	- Dominant sites of action include the genitourinary and gastrointestinal tracts - Side effects include excessive sweating, diarrhea, and abdominal cramps - Contraindications include asthma, COPD, and cardiovascular disease associated with bradycardia or hypotension
Dutasteride Finasteride	5 Alpha-reductase inhibitors	Antagonism of 5 alpha reductase receptor, thus blocking the conversion of testosterone to dihydrotestosterone	- Relieve obstruction of urinary outflow tract through reduction in prostatic mass - Clinical effect is slow, owing to the prostate’s deprivation of DHT - Side effects include impotence and gynecomastia

**Table 5 cancers-17-00614-t005:** Pharmacotherapy options for VTE prophylaxis in ERAS [29,30].

Drug	Class	Mechanism of Action	Aspects of Therapy Impacting ERAS
Fondaparinux Heparin Low-molecular-weight heparin (LMWH)	Anticoagulant heparin derivatives	Bind to antithrombin and catalyze the rate at which antithrombin is able to inhibit coagulation proteases	- Heparin is able to form a quaternary structure with antithrombin that allows for the inhibition of thrombin (factor II) in addition to factor X - Given that LMWH molecules are shorter in length, the binding with antithrombin does not always allow for the inhibition of factor II, so there is a preference for the inhibition of factor X at a 1:2–1:3 ratio - Must monitor for heparin-induced thrombocytopenia - LMWH is generally favored for its more consistent bioavailability and longer half-life - LMWH requires renal dose adjustments when CrCl falls below 30 mL/min

**Table 6 cancers-17-00614-t006:** Considerations for perioperative pharmacotherapy in select chronic medical conditions [29,30].

Chronic Condition	Pain	Nausea and Vomiting	Bladder Spasms	Urinary Retention	VTE Prophylaxis
Geriatric Patient	Opioids must be used with caution in geriatric patients due to increased susceptibility to sedation and delirium. Recommendation is to start with lower dose of NSAIDs due to increased risk of gastrointestinal or cardiovascular side effects. Gabapentin and pregabalin have been shown to cause somnolence, dizziness, ataxia, and fatigue, which may be exacerbated in the geriatric population. Geriatric patients’ nerves can be more sensitized to local anesthetics, thereby requiring lower doses.	5HT-3 receptor antagonists are well-tolerated, but dizziness is a common side effect that geriatric patients may be more susceptible to. Scopolamine must be used with caution in the geriatric patient population due to its anticholinergic side effect profile. Metoclopramide’s extrapyramidal side effect profile should be closely monitored in elderly patients, especially in those with Parkinson’s disease.	All antimuscarinic agents must be used cautiously in the geriatric patient population due to the risk of anticholinergic side effects. Darifenacin, solifenacin, and trospium are less effective at crossing the blood–brain barrier, reducing cognitive impairment risk. The use of antimuscarinics in elderly men with a history of BPH can further increase the risk of acute urinary retention.	Silodosin and tamsulosin have greater selectivity for the alpha-1a receptor. This selectivity reduces the risk of orthostatic hypotension when compared to alfuzosin, doxazosin, prazosin, and terazosin. However, orthostatic hypotension should be monitored throughout the class in the geriatric patient population.	In geriatric patients with a high risk of bleeding (such as due to a risk of falls), heparin and low-molecular-weight heparin should be used cautiously, as the potential for uncontrolled bleeding becomes more pronounced.
Heart Failure	NSAIDs can promote fluid retention, hypertension, and edema. This can potentially exacerbate heart failure. Notably, among over-the-counter NSAIDs, naproxen has a more favorable cardiovascular risk profile. Gabapentin and Pregabalin are known to induce peripheral edema, which may complicate heart failure management. Ketamine is able to stimulate the sympathetic nervous system, which can cause transient increases in blood pressure, heart rate, and cardiac output. It can also increase myocardial oxygen consumption.	Aprepitant can induce hypotension when used for postoperative nausea and vomiting. Exercise caution with the use of scopolamine in patients with preexisting heart failure, as anticholinergic action may exacerbate cardiac dysfunction. Metoclopramide has been demonstrated to increase cardiac risk. It should be avoided in patients with preexisting cardiac disease or risk factors.	In patients requiring muscarinic antagonists, agents with preference for the M3 receptor (darifenacin and solifenacin) are less likely to induce cardiac side effects. Mirabegron has been shown to have dose-related increases in blood pressure. In the setting of heart failure, this may require diligent monitoring.	The AHA has concluded that doxazosin, prazosin, tamsulosin, and terazosin may exacerbate underlying myocardial dysfunction. In patients with pronounced bradycardia or hypotension, the muscarinic effects of bethanechol may exacerbate these states.	No notable considerations for perioperative VTE prophylaxis in patients with heart failure.
Arrhythmia	Bupivacaine has been shown to be more cardiotoxic than equieffective doses of lidocaine. If the local anesthetic inadvertently enters the bloodstream, severe ventricular arrhythmias can be induced.	QT prolongation is a concern among all first-generation 5HT-3 receptor antagonists (dolasetron, granisetron, and ondansetron). Exercise caution with the use of scopolamine in patients with tachyarrhythmias, as anticholinergic action may exacerbate tachycardia. Metoclopramide has been known to cause sinus arrest as well as QT prolongation in at-risk patients.	Solifenacin and tolterodine should be used with caution in patients with a history of QT prolongation. Bupivacaine has been shown to be more cardiotoxic than equieffective doses of lidocaine. If the local anesthetic inadvertently enters the bloodstream, severe ventricular arrhythmias can be induced.	Alfuzosin has been shown to have a marginal effect on QT prolongation, which may be additive in the setting of other agents that prolong a patient’s QT interval.	Heparin and low-molecular-weight heparin can induce hyperkalemia through the suppression of aldosterone production. This hyperkalemia could potentiate arrhythmias.
COPD	Opioids can induce respiratory depression, which may be detrimental to patients with COPD and could induce acute respiratory failure. At high doses, fentanyl can induce chest wall rigidity, reducing ventilation. Ketamine acts as a bronchodilator, which may be beneficial in patients with reactive airways.	No notable considerations for perioperative nausea and vomiting management in patients with COPD.	No notable considerations for perioperative bladder spasm and stent discomfort management in patients with COPD.	While bethanechol is largely selective to the gastrointestinal and genitourinary tracts, bronchospasm can occur, which may contribute to acute respiratory failure in patients with COPD.	No notable considerations for perioperative VTE prophylaxis in patients with COPD.
Liver Disease	Opioids including fentanyl, hydrocodone, morphine, oxycodone, and tramadol all undergo hepatic metabolism. Impaired hepatic function may alter blood concentrations and elimination rates. NSAIDs are not recommended in advanced hepatic disease. Acetaminophen must be used cautiously in hepatic disease due to impaired metabolism. Additionally, its total daily dose must be monitored, as acetaminophen is commonly seen in combination with opioids. Patients with liver disease may have increased effects from local anesthetics unbound in the bloodstream.	Among the 5HT-3 receptor antagonists, ondansetron may require hepatic dose adjustment based on the degree of hepatic insufficiency.	Darifenacin, solifenacin, and tolterodine may require hepatic dose adjustment based on the degree of hepatic insufficiency. Patients with liver disease may have increased effects from local anesthetics unbound in the bloodstream. Mirabegron may require hepatic dose reduction depending on the degree of hepatic insufficiency.	Alfuzosin and silodosin may require hepatic dose adjustment based on the degree of hepatic insufficiency.	Heparin and low-molecular-weight heparin may induce reversible changes in liver enzyme function tests.
Diabetes	No notable considerations for perioperative pain management in patients with diabetes.	Dexamethasone, as well as other corticosteroids, can raise blood glucose levels. However, one-time dosing for PONV is unlikely to cause clinically significant hyperglycemia.	Mirabegron can increase the risk of developing a urinary tract infection. This risk may be further exacerbated in patients with diabetes due to preexisting glucosuria.	No notable considerations for perioperative urinary retention management in patients with diabetes.	No notable considerations for perioperative VTE prophylaxis in patients with diabetes.
CKD	Metabolism of morphine produces active metabolites. In patients with decreased renal function, the accumulation of active metabolites increases the risk of toxicity. If opioids are required, preference may be lent towards fentanyl, which has no active metabolites. NSAIDS should be used cautiously in CKD given their ability to induce AKI. Gabapentin and pregabalin may require renal dose adjustment based on degree of renal insufficiency.	Metoclopramide may require renal dose adjustment based on degree of renal insufficiency.	Solifenacin, tolterodine, and trospium may require renal dose adjustments based on the degree of renal insufficiency. Mirabegron may require renal dose adjustments based on the degree of renal insufficiency.	Silodosin may require renal dose adjustment based on degree of renal insufficiency.	Low-molecular-weight heparin may require renal dose adjustment based on degree of renal insufficiency.

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
