# Peer review of "Optimizing Pharmacotherapy During Implementation of Enhanced Recovery After Surgery (ERAS) in Ambulatory Urologic Oncology Surgery: Narrative Review"

_cancers, 2025, doi:10.3390/cancers17040614_

Round 1
Reviewer 1 Report
Comments and Suggestions for Authors
This narrative review examines various aspects of pharmacological therapy within the ERAS Protocol. A section-by-section analysis reveals the following:
- Introduction: The introduction is engaging and aligns well with the title.
- Methods: The methodology entirely lacks details on inclusion and exclusion criteria as well as the literature search process.
- Results: The results section omits a literature screening and lacks descriptions of study characteristics.
- Discussion: The discussion addresses all major points; however, several issues remain:
- In Section on Artificial Intelligence and ERAS, while the sole study identified is intriguing, the authors exhibit redundancy and fail to emphasize the need for further exploration to establish the definitive parameters for an algorithm.
- The section on DVT Prophylaxis appears incomplete, as it does not address the timing and dosage of prophylactic heparin.
Grammar and Style: In some areas, the text is overly verbose, which detracts from its suitability for an academic paper. A more concise and formal revision of the text is recommended.
Comments on the Quality of English LanguageEnglish is appropriate but not adapted for scientific paper
Reviewer 2 Report
Comments and Suggestions for Authors
This manuscript provides a comprehensive review of the implementation of Enhanced Recovery After Surgery (ERAS) protocols with a focus on pharmacotherapy in urologic oncology procedures. The topic is highly relevant. The work is well-written and organized, with clear headings and logical flow. It appropriately incorporates supporting evidence and provides actionable insights for practitioners. However, there are areas where the manuscript could benefit from refinement to enhance its impact and clarity:
-
While the literature search and focus are clear, providing more details on search strategies (e.g., inclusion/exclusion criteria, date range) would enhance transparency and reproducibility.
-
The section on artificial intelligence (AI) and ERAS protocols is promising but could benefit from specific examples of clinical implementation or limitations. Discuss potential barriers to adoption, such as cost or clinician training, to provide a balanced view.
-
Tables summarizing pharmacotherapy options are useful but could include additional details, such as comparative efficacy or specific clinical scenarios where each option is preferred.
-
While the work is comprehensive, some claims could be bolstered with more recent or high-level evidence. For instance, emerging data on the cost-effectiveness of ERAS in specific procedures could further validate the recommendations.
-
The narrative is clear for an expert audience but might benefit from a simplified summary or visual aids to support comprehension for multidisciplinary readers, including non-specialists.
-
While the paper provides a robust analysis of current ERAS strategies, explicitly outlining gaps in knowledge or potential areas for future investigation (e.g., immunonutrition or opioid-sparing approaches) would add value.
Round 2
Reviewer 1 Report
Comments and Suggestions for Authors
The authors have successfully addressed all of my concerns, and I have no additional comments to provide.